# microRNAs Associated with Gemcitabine Resistance via EMT, TME, and Drug Metabolism in Pancreatic Cancer

**DOI:** 10.3390/cancers15041230

**Published:** 2023-02-15

**Authors:** Naotake Funamizu, Masahiko Honjo, Kei Tamura, Katsunori Sakamoto, Kohei Ogawa, Yasutsugu Takada

**Affiliations:** Department of Hepatobiliary Pancreatic and Transplantation Surgery, Ehime University Graduate School of Medicine, 454 Shitsukawa, Toon 791-0295, Ehime, Japan

**Keywords:** microRNA, gemcitabine, pancreatic cancer, biomarker, chemoresistance

## Abstract

**Simple Summary:**

We herein reviewed the current evidence for the role of microRNAs (miRNAs) in the mechanism of chemoresistance in pancreatic cancer. Pancreatic cancer has an extremely poor prognosis due to its late discovery, aggressive nature, and chemoresistance. Recent accumulated reports proved that aberrant miRNAs could induce chemoresistance in pancreatic cancer. However, the exact underlying molecular mechanisms remain poorly understood. In this review, we discuss recently available and novel knowledge about overcoming chemoresistance in pancreatic cancer.

**Abstract:**

Despite extensive research, pancreatic cancer remains a lethal disease with an extremely poor prognosis. The difficulty in early detection and chemoresistance to therapeutic agents are major clinical concerns. To improve prognosis, novel biomarkers, and therapeutic strategies for chemoresistance are urgently needed. microRNAs (miRNAs) play important roles in the development, progression, and metastasis of several cancers. During the last few decades, the association between pancreatic cancer and miRNAs has been extensively elucidated, with several miRNAs found to be correlated with patient prognosis. Moreover, recent evidence has revealed that miRNAs are intimately involved in gemcitabine sensitivity and resistance through epithelial-to-mesenchymal transition, the tumor microenvironment, and drug metabolism. Gemcitabine is the gold standard drug for pancreatic cancer treatment, but gemcitabine resistance develops easily after chemotherapy initiation. Therefore, in this review, we summarize the gemcitabine resistance mechanisms associated with aberrantly expressed miRNAs in pancreatic cancer, especially focusing on the mechanisms associated with epithelial-to-mesenchymal transition, the tumor microenvironment, and metabolism. This novel evidence of gemcitabine resistance will drive further research to elucidate the mechanisms of chemoresistance and improve patient outcomes.

## 1. Introduction

Pancreatic cancer remains a lethal disease and is the third leading cause of cancer-related deaths in the United States, with an incidence of 62,210 new cases, and an exceptionally high mortality rate of 49,830 deaths (80.1%) in 2022 [1]. Pancreatic cancer is expected to become the second leading cause of cancer-related deaths in the future [2]. Despite accumulated knowledge regarding pancreatic cancer etiology, patient prognosis has not significantly improved over the last decade [3]. The high mortality rate is rooted in the lack of specific symptoms, diagnostic tools, and effective chemotherapeutics, along with the increasing incidence of chemoresistance. Among these, chemoresistance is a well-known factor that contributes to a poor prognosis.

Compared with 5-fluorouracil, gemcitabine was found to be superior in relieving symptoms caused by progression in patients with pancreatic cancer [4]. This made gemcitabine the key drug for pancreatic cancer in combination with other chemotherapeutic agents such as nab-paclitaxel [5] or oxaliplatin [6]. However, the clinical benefits of these regimens are limited by chemoresistance. Thus, there is an urgent and crucial need to establish strategies for overcoming gemcitabine resistance to improve pancreatic cancer treatment. To date, the most clinically reliable marker for prognosis is CA19-9) (carbohydrate antigen 19-9 in pancreatic cancer [7]. Several other markers have been reported; however, they have not seen clinical use [8,9,10].

microRNAs (miRNAs) regulate gene expression by binding to the 3′-UTR (prime untranslated region) of their target mRNAs and inhibiting the production of those proteins. The association between miRNAs and chemoresistance has been reported in various types of cancer [11,12,13]. Thus, miRNAs are attractive molecules that retain the potential to overcome gemcitabine resistance. However, the scientific mechanism underlying this association remains unresolved despite extensive study. Recent evidence has shown that epithelial-to-mesenchymal transition (EMT) [14], tumor microenvironment (TME) [15,16], and drug metabolism [17] are major causes of the development of gemcitabine resistance via aberrantly expressed miRNAs. Therefore, it is important to identify innovative biomarkers and novel targets for intrinsic and acquired gemcitabine resistance in patients with pancreatic cancer, since CA19-9 is not a biomarker for gemcitabine resistance, though it is known to be a prognostic marker with high sensitivity and specificity (both 80%) [7]. EMT is a process that changes epithelial cells into mesenchymal-like cells, leading to the acquisition of cellular properties for invasiveness, proliferation, chemoresistance, and higher metastatic potential. Evidence has shown that expression of EMT-related genes such as those encoding vimentin, Snai1, and Zeb1 (zinc finger E-box binding homeobox 1) could further increase the risk of cancer cells acquiring a high-grade phenotype [18,19].

The TME is composed of interstitial tissues surrounding the pancreatic cancer cells and comprises immune cells, pancreatic stellate cells, stromal cells, blood vessels, fibroblasts, and the extracellular matrix (ECM). The TME can induce hypoxia, which confers several metabolic advantages for cancer cell protection and survival [20,21]. Recent review articles have shown that TME plays a pivotal role in the development of pancreatic cancer cells and gemcitabine resistance [22,23]. 

Finally, aberrant gemcitabine metabolism, including gemcitabine-related transporters and enzymes, is a cause of gemcitabine resistance [24,25,26,27]. Dysregulation of these transporters or enzymes results in gemcitabine resistance through complex, irregular factors and signaling pathways [28]. 

Based on these reports, the purpose of this review was to summarize the current evidence on the roles of miRNAs as potential biomarkers and therapeutic targets for gemcitabine resistance via EMT, TME, and drug metabolism in pancreatic cancer. Here, we systematically summarized recent published reports, including review articles, and focused on gemcitabine resistance in pancreatic cancer. A systematic search was performed using PubMed to identify original research studies mainly published in the last 20 years using the search terms pancreatic cancer and gemcitabine resistance. Finally, 1357 articles were chosen. Moreover, we targeted studies that satisfied the following criteria: (1) studied miRNA expression due to gemcitabine resistance in pancreatic cancer, (2) used human samples, and (3) were written in English. The full texts of the articles of interest were then evaluated. In this review, we focused on the mechanisms of gemcitabine resistance via EMT, TME, and gemcitabine metabolism. 

## 2. Diverse Functions of miRNAs

miRNAs are short noncoding RNAs containing approximately 22 nucleotides, which can bind directly to specific targets within the 3′-untranslated regions of mRNAs [29,30]. In addition, miRNAs inhibit translation or enhance mRNA cleavage at the post-transcriptional level [31]. miRNAs are key regulators in the development, differentiation, and apoptosis of normal cells and may affect tumorigenesis and metastatic potential [32]. Remarkably, miRNAs exhibit tissue-specific and disease-specific expression, which suggests their potential as novel diagnostic and prognostic markers as well as therapeutic targets [33]. In addition, each miRNA has several different target genes, which results in the inhibition of the target genes for signaling pathways; therefore, miRNAs can function as both oncogenes and tumor suppressor genes [34]. Recent data have shown that frequently deregulated miRNAs are associated with detection, progression, and chemoresistance in pancreatic cancer [35,36]. Furthermore, several studies have documented that miRNAs influence gemcitabine resistance via EMT, TME, and drug metabolism [37,38,39]. 

## 3. Gemcitabine Resistance in Pancreatic Cancer 

Gemcitabine remains the gold standard for the treatment of pancreatic cancer. Clinical studies, including combination treatment with gemcitabine, have been extensively performed; however, combination regimens were found to be insufficient compared with traditional regimens, such as gemcitabine plus nab-paclitaxel or erlotinib, or S-1 combination [40,41,42,43]. The major limitations in treatment effects include gemcitabine resistance and considerable adverse events [28]. An understanding of the mechanisms underlying gemcitabine resistance may lead to the discovery of promising targeted therapies. Numerous studies have reported the major molecular factors of gemcitabine resistance, such as: (1) cellular factors (cell signaling pathways, cancer stem cells, and EMT); (2) environmental factors (oxygen supply, stroma, ECM, and blood supply); and (3) drug metabolism (transporter, drug activation, and enzymes) in pancreatic cancer [22,44,45,46]. Moreover, these factors can result from molecular and cellular changes, including aberrant expression of mRNAs or miRNAs, and can interact in a complex manner [47]. It is therefore challenging to overcome gemcitabine resistance given the several different factors involved. Herein, we summarize the reported miRNA-mediated mechanisms that directly contribute to gemcitabine sensitivity and resistance in pancreatic cancer. Among them, we focus on EMT, TME, and metabolism-related mechanisms via miRNAs.

## 4. EMT-Mediated Mechanisms of Gemcitabine Resistance in Pancreatic Cancer

### 4.1. EMT and Gemcitabine Resistance

EMT is associated with gemcitabine resistance in pancreatic cancer [48,49]. It is characterized by the evolution of an epithelial phenotype into a mesenchymal phenotype, which can lead to cell proliferation, invasiveness, and metastasis [50,51]. This fundamental process is accompanied by morphological changes in the cancer cells. EMT is mediated by a variety of key genes and cellular signaling pathways; consequently, EMT results in higher proliferation, invasion, and chemoresistance. Molecular markers of EMT include increased expression of vimentin, Twist, Snail, Slug, and ZEB1 [52,53]. In contrast, mesenchymal-epithelial transition (MET) markers include Zo-1 and E-cadherin. Signaling pathways, such as the Notch and NFκB pathways, are critical for the induction of EMT [50]. Several studies have reported that gemcitabine-resistant cells show upregulated vimentin and downregulated E-cadherin expression associated with the activation of NFkB and c-MET tyrosine kinase, respectively [54,55,56]. Additionally, gemcitabine-resistant cells express higher SMAD2 or cancer stem cell (CSC) markers with EMT characteristics [57,58], because CSCs are evidently linked to the EMT process, which has been associated with gemcitabine resistance [59]. miR-34a recovers sensitivity to gemcitabine by inhibiting Notch 1, which is located upstream of the EMT pathways [60]. EMT can directly induce the CSC phenotype in pancreatic cancer. In contrast, gemcitabine induces EMT and CSC molecular marker expression [61]. Thus, EMT features overlap with molecular and morphological changes in CSC. In contrast, miR-17 is reduced in gemcitabine-resistant CSC by targeting the TGF-β1 signaling pathway and inhibiting the downstream targets p21, p57, and TBX3 [62]. Moreover, Ji et al. revealed that miR-34 may be involved in pancreatic cancer stem cell self-renewal, potentially via the direct modulation of downstream targets Bcl-2 and Notch [63]. In this review, we considered CSCs to be consistent with the EMT phenomenon. Conversely, inhibition of Notch signaling in gemcitabine-resistant cells could induce the MET phenotype from the EMT phenotype, indicating that gemcitabine resistance is reversible and associated with decreased EMT marker expression [58]. Hypoxia can also induce PLOD2-influenced gemcitabine resistance through EMT [64]. Moreover, ZEB1 can also mediate gemcitabine resistance and reduce E-cadherin expression. In contrast, reduced ZEB1 levels can restore gemcitabine sensitivity, indicating that ZEB1 is responsible for gemcitabine resistance [65]. Carrasco-Garcia et al. also showed that SOX9, which participates in the initiation of pancreatic cancer, correlated with EMT with high vimentin and low E-cadherin expression [66]. Recent evidence has shown that signaling pathways, including tumor necrosis factor (TNF) and hypoxia-inducible factor-1 (HIF-1), are associated with EMT in pancreatic cancer [66,67]. Moreover, Zhang et al. showed that DPEP1 induction enhanced gemcitabine sensitivity through the RAS-RAF-MEK-ERK and PI3K pathways [68]. EMT is also associated with tumor budding, which accounts for gemcitabine resistance; a study indicated that tumor budding with vimentin expression becomes a key process in pancreatic cancer and is responsible for progression and gemcitabine resistance [69]. Taken together, EMT induction is strongly correlated with the development of gemcitabine resistance in pancreatic cancer. 

### 4.2. miRNAs Related to Gemcitabine and EMT

#### 4.2.1. miRNAs Associated with Gemcitabine Sensitivity

With regard to the association between gemcitabine sensitivity and miRNAs mediating EMT, several tumor suppressor miRNAs have been identified to regulate EMT-related genes and improve gemcitabine sensitivity in pancreatic cancer [70,71]. (See Figure 1 and Table 1). Wang et al. showed that miR-30a reverses gemcitabine resistance in pancreatic cancer by targeting the Snail-AKT signaling pathway [72]. Funamizu et al. revealed that miR-200b could restore gemcitabine sensitivity by inhibiting ZEB1, thereby upregulating E-cadherin [73]. In addition, Li et al. proved that naturally occurring agent-induced miR-200 and let-7 expression could reverse MET from the EMT phenotype in gemcitabine-resistant cells [74]. Liu et al. reported that miR-125a-3p can restore gemcitabine sensitivity and inhibit EMT through targeting Fyn [75]. Fu et al. also revealed that NEAT1, mediated by miR-506, could control ZEB2 expression [76]. Additionally, Li et al. showed that miR-506 suppressed sphingosine kinase 1, which was significantly associated with poor survival in a large cohort [77]. Hiramoto et al. demonstrated that miR-509 and miR-1243 improve gemcitabine sensitivity via E-cadherin expression [78]. Furthermore, Yang et al. reported that miR-3656 plays a significant role in gemcitabine sensitivity by inhibiting vimentin and Twist expression [79]. Two reports have shown that miR-153 can enhance gemcitabine sensitivity by inhibiting Snail [80,81]. Wang et al. revealed that upregulation of miR-183 and miR-200b improved gemcitabine sensitivity via ZEB1 inhibition caused by KLF4 [82]. Chaudhary et al. also reported that miR-205 impairs gemcitabine resistance by inhibiting the EMT phenotype [83]. More recently, Yang et al. revealed that exosomal miR-210 in CSCs mediates gemcitabine resistance by activating the PI3K/Akt/mTOR pathway [84].

#### 4.2.2. miRNAs Associated with Gemcitabine Resistance

The reported miRNAs associated with gemcitabine resistance in pancreatic cancer have been summarized. Hasegawa et al. revealed that miR-1246 contributes to gemcitabine resistance and induces CSC-like properties through CCNG2 [85]. Xiong et al. demonstrated that miR-10a contributes to gemcitabine resistance by targeting TFAP2C, thereby resulting in increased Snail 1 expression and EMT induction [86]. Zhang et al. showed that miR-15b degrades SMURF2 and promotes TGF-β-mediated EMT in pancreatic cancer [87]. In addition, Yang and Funamizu et al. demonstrated that miR-301b induced EMT and enhanced gemcitabine resistance by reducing E-cadherin expression [54,88]. Zhang et al. also showed that activation of the miR-301/TP63 axis caused by hypoxia-induced EMT contributes to gemcitabine resistance [89]. Okazaki et al. suggested that miR-296-5p induces EMT and gemcitabine resistance [90]. Meanwhile, Ma et al. demonstrated that miR-223 enhances gemcitabine resistance by targeting FBXW7 through activating Notch signaling-mediated EMT [91]. Yu et al. revealed that miR-1206 enhances gemcitabine resistance through ESRP1 inhibition [92]. Furthermore, Yu et al. showed that miR-497 could inhibit gemcitabine resistance in CSCs by targeting NFκB [93]. Ma et al. indicated that miR-200-3p attenuates gemcitabine resistance in CSCs by modulating EMT and stemness [94].

**Table 1 cancers-15-01230-t001:** miRNAs for EMT.

Author	Ref. Number	miRNA	Target Gene
Sensitivity			
Li Y	[74]	let-7	NA
Cioffi M	[62]	miR-17	*NODAL*
Wang T	[72]	miR-30a	*SNAI1*
Ji Q	[63]	miR-34	*Bcl-2*
Liu G	[75]	miR-125a	*Fyn*
Liu F	[80]	miR-153	*Snail*
Bai Z	[81]	miR-153	*Snail*
Wang Z	[82]	miR-183	*ZEB1*
Li Y	[74]	miR-200	NA
Funamizu N	[55]	miR-200b	*ZEB1*
Wang Z	[82]	miR-200b	*ZEB1*
Ma C	[94]	miR-200c	NA
Chaudhary AK	[83]	miR-205	*TUBB3*
Yu Q	[93]	miR-497	*NFKB1*
Fu X	[76]	miR-506	*NEAT1*
Hiramoto H	[78]	miR-509	NA
Yu S	[92]	miR-1206	*ESRP1*
Hiramoto H	[78]	miR-1243	NA
Yang RM	[79]	miR-3656	EMT related genes
Resistance			
Xiong G	[86]	miR-10a	*TFAP2C*
Zhang WL	[87]	miR-15b	*SMURF2*
Yang Z	[84]	miR-210	*Rapamycin*
Ma J	[91]	miR-223	*FBXW7*
Okazaki J	[90]	miR-296	*BOK*
Zhang KD	[89]	miR-301	*TP63*
Funamizu N	[54]	miR-301b	*TP63*
Yang S	[88]	miR-301b	*NR3C2*
Hasegawa S	[85]	miR-1246	*CyclinG2*

Ref: reference, NA: not applicable.

Despite this evidence, the role of miRNAs in gemcitabine sensitivity and resistance remains controversial because the complex and interacting networks underlying the phenomenon are challenging to elucidate. However, accumulating evidence implicates that aberrant expression of miRNAs modulates the responsiveness of gemcitabine sensitization.

## 5. TME-Mediated Mechanisms of Gemcitabine Resistance in Pancreatic Cancer

### 5.1. TME and Gemcitabine Resistance in Pancreatic Cancer

Pancreatic cancer has unique features to survive therapeutic strategies, characterized by the presence of an extensive desmoplastic stroma composed of ECM, cancer-associated fibroblasts (CAFs), pancreatic stellate cells (PSCs), inflammatory cells, immune cells (including tumor-associated macrophages [TAMs]), and other cell types (such as endothelial cells). The dense stroma facilitates the compression of blood vessels and leads to a hypoxic environment, which reduces the supply of chemotherapeutic agents and supports cancer progression [95]. Under these circumstances, the TME can induce immunosuppression to escape the immune system. The desmoplastic stroma and hypoxic environment have also been reported to promote EMT and cause gemcitabine resistance [96,97]. Accumulated evidence has revealed that desmoplastic stroma simply does not develop a physical barrier to gemcitabine; moreover, the respective components in the TME act to resist gemcitabine [98,99,100,101].

### 5.2. Role of ECM in Gemcitabine Resistance

ECM is defined as the physical support and material that fills the extracellular space and functions as a scaffold for cell adhesion, including fibronectin, collagen, and proteoglycan. ECM components, including collagen, fibronectin, and laminin, are secreted by PSCs and CAFs [95]. Recent evidence has shown that the ECM functions as a supporting tissue, regulating cancer cell proliferation and EMT [102]. Fibronectin is a major component of the ECM that Miyamoto et al. have demonstrated to be a contributor to gemcitabine resistance [103]. In addition, fibronectin plays a key role in gemcitabine resistance by activating the ERK1/2 pathway [104]. Topalovski et al. indicated that cooperation between TGF-β and fibronectin may establish coordinated EMT induction [105]. Furthermore, fibronectin can support cancer progression and reduce gemcitabine response [105]. In contrast, Dangi-Garimella et al. demonstrated that membrane type 1 matrix metalloproteinase (MMP) contributes to gemcitabine resistance and suggested that targeting MMP could be a novel approach to improving gemcitabine sensitivity [106]. Thus, ECM-targeted agents combined with gemcitabine chemotherapy are promising strategies for overcoming gemcitabine resistance. However, two clinical studies involving anti-ECM inhibitors did not show significant efficacy of these drugs [107,108].

### 5.3. Role of CAFs in Gemcitabine Resistance

CAFs constitute a major part of the tumor mass in pancreatic cancer. CAFs can enhance chemoresistance through ECM remodeling and immunological reprogramming [109]. Richards et al. recently reported that CAFs have the potential for gemcitabine resistance. They also showed that exosomes produced by CAFs promote EMT and gemcitabine resistance by inducing Snail [110]. In addition, Zhang et al. showed that CAFs activate NFκB and IL1 receptor-associated kinase 4, which results in enhanced tumor fibrosis, cell proliferation, and gemcitabine resistance [111]. Recent reports have revealed that CAFs promote gemcitabine resistance via the LIF/STAT3 or the TGF-β1/SMAD2/3 pathway [112,113], suggesting that targeting these pathways may be a novel strategy to reverse gemcitabine resistance. Thus, therapeutic agents controlling CAF function may play a vital role in sensitizing gemcitabine.

### 5.4. Role of PSCs in Gemcitabine Resistance

PSCs provide a solid foundation for the production of collagenous stroma for cancer development and survival [114,115]. Unfortunately, the role of PSCs in gemcitabine resistance has not yet been fully elucidated. However, recent evidence suggests that PSCs act as drivers of gemcitabine resistance and cancer progression. Cao et al. showed that the Notch pathway activated by PSCs promotes gemcitabine resistance and induces EMT [116]. Interestingly, PSCs have no tolerance to glucose adjustment; therefore, an increased number of PSCs in the pancreas allows the development of type 2 diabetes. Moreover, EMT-mediated higher glucose levels promote malignant potential [117]. It is well known that PSCs promote EMT in pancreatic cancer cells [118]. These reports suggest that targeting the Notch pathway may be an effective strategy for recovering gemcitabine tolerance. 

### 5.5. Role of TAMs in Gemcitabine Resistance

TAMs are also abundantly present in the TME. TAMs innately play a role in the phagocytosis of apoptotic cells and affect cancer progression and gemcitabine resistance by secreting numerous factors, such as growth factors, proteolytic enzymes, and inflammatory cytokines [119,120]. Weiseman et al. revealed that TAMs contribute to gemcitabine resistance by reducing apoptosis and upregulating cytidine deaminase (CDA) expression. These effects augment the response to gemcitabine by activating caspase-3 [121]. Moreover, Guo et al. showed that miR-222, delivered by TAM, suppresses TSC1 and activates the PI3K/AKT/mTOR pathway, which results in the development of gemcitabine-resistant cancer cells [122]. Additionally, Nagathihalli et al. showed that a STAT3 pathway inhibitor combined with gemcitabine can enhance gemcitabine delivery and response by remodeling the tumor stroma [123]. In contrast, TAMs directly activate the STAT3 pathway to regulate CSCs [124]. Thus, STAT3-targeted therapy with gemcitabine may be a promising therapeutic strategy for pancreatic cancer.

### 5.6. miRNAs Involved in TME-Mediated Gemcitabine Resistance 

Hypoxia is an essential component of the TME for the survival of cancer cells in pancreatic cancer [125,126,127]. Extensive studies have revealed that miRNAs are strongly associated with TME function [128,129,130]. (Figure 2 and Table 2). Luo et al. revealed that miR-301a plays a critical role in gemcitabine resistance via the TME [131]. Liu et al. also showed that PVT1 and HIF-1 inhibition, mediated by miR-143, improves gemcitabine sensitivity [132]. Xin et al. proposed that nano-medically modified gemcitabine and miR-519c are effective therapeutic targets to treat desmoplasia and hypoxia-induced gemcitabine resistance in the TME [133]. In contrast, HIF-1 expression is a well-known inducible factor for gemcitabine resistance and EMT [134,135]. As such, HIF-1-targeted miRNA strategies have been described in the literature. Liu et al. showed that miR-3662 inhibits gemcitabine resistance by inhibiting HIF-1 in the TME [136]. In addition, Ni et al. showed that miR-210 controls HOXA9 expression and upregulates the HIF-1/NF-κB pathway, which promotes EMT and hypoxia [137].

A recent report showed that miR-21 affects CAFs accumulation, and subsequently, CAFs release miR-21 that can induce gemcitabine resistance by downregulating PTEN [111,138,139]. Notably, hypoxia induces miR-21 expression [115]. Moreover, tumor associated fibrosis produces miR-21 that targets PTEN [140]. miR-21 is a powerful oncogenic gene that enhances gemcitabine resistance by targeting PTEN or FasL [141]. In contrast, Park et al. reported that miR-21 inhibition or miR-221 induction enhanced gemcitabine sensitivity in pancreatic cancer due to increased PTEN expression [142]. Fang et al. showed that CAF-derived exosomal miR-106b plays a significant role in gemcitabine resistance [143]. Thus, miR-21-mediated CAF-targeted therapy may be a promising strategy to overcome chemoresistance in TME.

**Table 2 cancers-15-01230-t002:** miRNAs for TME.

Author	Ref. Number	miRNA	Target Gene
Sensitivity			
Liu YF	[132]	miR-143	*HIF-1*
Ni J	[137]	miR-210	*HOXA9*
Liu A	[136]	miR-3662	*HIF-1*
Resistance			
Zhang L	[138]	miR-21	*PDCD4*
Kadera BE	[139]	miR-21	NA
Fang Y	[143]	miR-106b	*TP53INP1*
Masamune A	[144]	miR-221	NA
Guo Y	[122]	miR-222	*TSC1*
Luo G	[131]	miR-301a	*TP63*
Xin X	[133]	miR-519c	*HIF-1*

Ref: reference, NA: not applicable.

Finally, increased miR-221 expression in PSCs contributes to gemcitabine resistance by activating the MAPK signaling pathway [144]. These findings suggest the importance of miRNAs in the regulation of TME components and their potential role as novel targets for improving gemcitabine efficacy in pancreatic cancer.

## 6. Gemcitabine Metabolism-Mediated Mechanisms of Gemcitabine Resistance in Pancreatic Cancer

### 6.1. Gemcitabine Metabolism

Previously, numerous investigations regarding the transport- and metabolism-related genes for gemcitabine were performed to elucidate the mechanism of gemcitabine resistance in pancreatic cancer cells [145,146,147]. Gemcitabine (dFdC) is a deoxycytidine nucleoside analog. Its metabolic pathways are shown in Figure 3. Gemcitabine is transported into the cells by nucleoside transporters, including concentrative nucleoside transporters (hCNTs) and equilibrative nucleoside transporters (hENT1 and hENT2). hENT1 is the main transporter for gemcitabine uptake in cancer cells. Gemcitabine is a prodrug that requires intracellular phosphorylation for its activation. dFdC is first phosphorylated to dFdCMP by deoxycytidine kinase (dCK), a key enzyme in the rate-limiting process. Subsequently, double phosphorylation results in the formation of the triphosphate form (dFdCTP). Finally, dFdCTP is incorporated into the DNA chain to produce a therapeutic effect. However, most dFdCs are inactivated by CDA. Additionally, dFdCDP and dFdCTP can inhibit ribonucleotide reductase (RR), which is responsible for converting ribonucleosides to deoxyribonucleoside triphosphates (dNTPs). Moreover, the dCTP produced by RR acts as a competitive inhibitor of dFdCTP. Therefore, overt changes in gemcitabine metabolism may generate gemcitabine resistance [148,149,150,151,152,153].

### 6.2. Relation of Gemcitabine Metabolism to Gene Expression in Pancreatic Cancer

#### 6.2.1. Transporters Associated with Gemcitabine Resistance

Alterations in hENT1, the major transporter for gemcitabine [153], contribute to the development of gemcitabine resistance [154]. Tsesmetzis et al. summarized that nucleoside analogs are principally transported by two membrane transporter families: hCNT1-3 and hENT1-4. Nucleoside analogs are transported by both hENTs and hCNTs, whereas nucleoside analogs are excreted by multidrug resistance proteins (MRP), also called ATP-binding cassette transporters (ABC transporters), which are classified into seven different subtypes (from ABCA- to ABCG) based on their gene structure [155]. Giovannetti et al. reported an association between hENT1 and gemcitabine sensitivity [156]. Mackey et al. also reported that hENT1 deficiency causes significant gemcitabine resistance [157]. In addition, recent data have shown that highly expressed hENT1 is a molecular and mechanistically relevant predictive marker for gemcitabine sensitivity [158,159]. In contrast, Hung et al. revealed that hCNT1 deficiency affects gemcitabine resistance [150]. Moreover, Skrypek et al. showed that MUC4-induced hcNT1 inhibition enhances gemcitabine resistance [160]. Qiu et al. described that inhibition of the PI3K/NFκB pathway could induce ABC transporter reduction by improving gemcitabine resistance [161]. Hagmann et al. also showed that ABC transporters are associated with gemcitabine resistance [149]. Because of the key roles played by hNT and ABC transporters in gemcitabine resistance, modulation of these transporters may lead to better efficacy of gemcitabine-based chemotherapeutic strategies.

#### 6.2.2. CDA-Induced Gemcitabine Resistance

CDA is a catabolic enzyme that transforms gemcitabine into an inactivated metabolite [162]. Several studies have shown that the upregulation of CDA confers gemcitabine resistance and gemcitabine-induced toxicities [24,163,164,165,166]. Moreover, CDA expression levels can be used as a predictive marker for gemcitabine resistance [167]. Therefore, upregulated CDA expression may play an important role in gemcitabine resistance and patient prognosis in pancreatic cancer.

#### 6.2.3. Relation of dCK and Gemcitabine Resistance

Gemcitabine is phosphorylated by dCK, resulting in its active product. dCK is the rate-limiting enzyme for nucleoside analogs [168,169]. Therefore, dCK inhibition is considered one of the causes of the development of gemcitabine resistance [25,170]. Ohmime et al. suggested that dCK expression is a prognostic factor in patients with pancreatic cancer [171]. Kamada et al. also reported that transduction of dCK could recover gemcitabine sensitivity in pancreatic cancer [172].

#### 6.2.4. Relation of RR to Gemcitabine Resistance

Another potential factor in the mechanism for gemcitabine resistance is the overexpression of RR. RR is composed of the regulatory subunit M1 and the catalytic subunit M2, which are responsible for the conversion of ribonucleosides to deoxyribonucleoside triphosphates [173]. Duxbury et al. found that upregulated RRM2 closely participates in gemcitabine resistance by inhibiting the NF-κB signaling pathway [174]. Furthermore, previous studies have shown that increased RR is involved in the enhanced resistance to gemcitabine [24,175]. In contrast, Nakahira et al. revealed that RRM1 inhibition significantly reduces gemcitabine resistance [27]. Moreover, several studies have revealed that increased RR expression serves as a prognostic marker for patients with bile duct and pancreatic cancers [176,177].

### 6.3. Relation of Gemcitabine Metabolism to miRNAs

Amponsah et al. recently clarified the impact of miRNAs on gemcitabine metabolism, including transporters and gemcitabine-related enzymes. This study showed that miR-210 is responsible for gemcitabine sensitivity by targeting the ABCC5 gene [178]. Gu et al. indicated that miR-3178 promotes chemoresistance to gemcitabine by upregulating the ABC transporter-mediated RhoB/PI3K/Akt pathway [179]. Although Wang et al. did not mention gemcitabine resistance, they reported an association between miR-520h and ABCG2 [180]. Additionally, miR-93 enhances ABCB1 expression via targeting PTEN and increases Akt phosphorylation [181]. Furthermore, miR-331 induces ABCB1 expression by activating the Wnt/β-catenin pathway through ST7L in pancreatic cancer cells [182]. To the best of our knowledge, there are no reports regarding the association of hENT and hCNT with miR-mediated gemcitabine resistance. Recent data have shown that miR-101-3p and miR-211 enhance gemcitabine sensitivity by inhibiting RRM1 and RRM2, respectively [17,183]. Bhutia et al. revealed that let-7 negatively regulates RRM2 and sensitizes to gemcitabine [184]. Moreover, Rajabpour et al. showed that miR-608 leads to increased gemcitabine sensitivity, with decreased RRM1 and CDA expression [185]. Lu et al. suggested that miR-20a-5p upregulates gemcitabine sensitivity by targeting RRM2 [186]. Patel et al. also recently demonstrated that suppression of miR-155 induced dCK levels and restored gemcitabine sensitivity [187]. This demonstrates that miRNAs are connected with numerous targets involved in resistance to gemcitabine metabolism (Figure 3 and Table 3). 

## 7. Conclusions

We believe gemcitabine plus miRNA-based therapeutics can be expected to overcome the poor prognosis of pancreatic cancer, although further studies are needed to elucidate the molecular mechanism of chemoresistance caused by aberrantly expressed miRNAs. One of the major limitations of miRNA-based therapeutics is the lack of a delivery system capable of targeting tumors. Recent clinical trials of miR-34a liposomal-based therapy have been performed for advanced solid cancers [188]. Unfortunately, the trial was halted due to serious adverse events. However, miRNA-based strategies have the potential to deliver unprecedented value in pancreatic cancer treatment, as these may be able to control several target genes and signaling pathways. Moreover, recent studies showed that secreted microvesicles such as exosomes contain miRNAs, which play an important role in gemcitabine chemoresistance [110,189,190]. Comandatore et al. reviewed that exosomes, including oncogenic miRNAs, can affect gemcitabine resistance [191]. Thus, we suggest that miRNA-based strategies, involving regulation of miRNA expression, might contribute to overcoming gemcitabine resistance in the future. In contrast, exosomes, including specific miRNAs, must be a biomarker for gemcitabine resistance due to liquid biopsy.

## Figures and Tables

**Figure 1 cancers-15-01230-f001:**
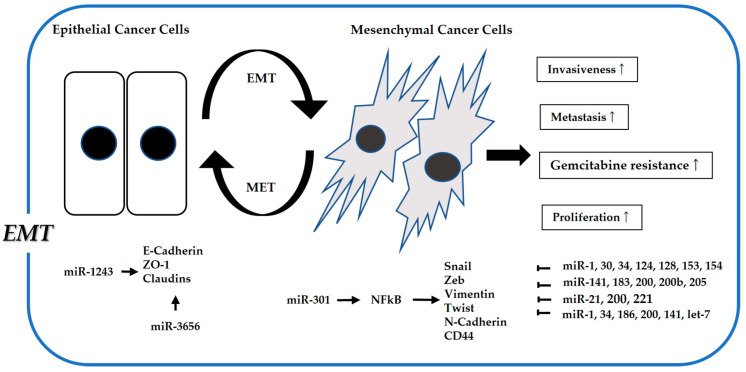
Association between microRNAs related to EMT (Epithelial–mesenchymal transition) and gemcitabine resistance. EMT is well known to increase drug resistance, cell proliferation, and invasiveness through acquiring properties like cancer stem cells.

**Figure 2 cancers-15-01230-f002:**
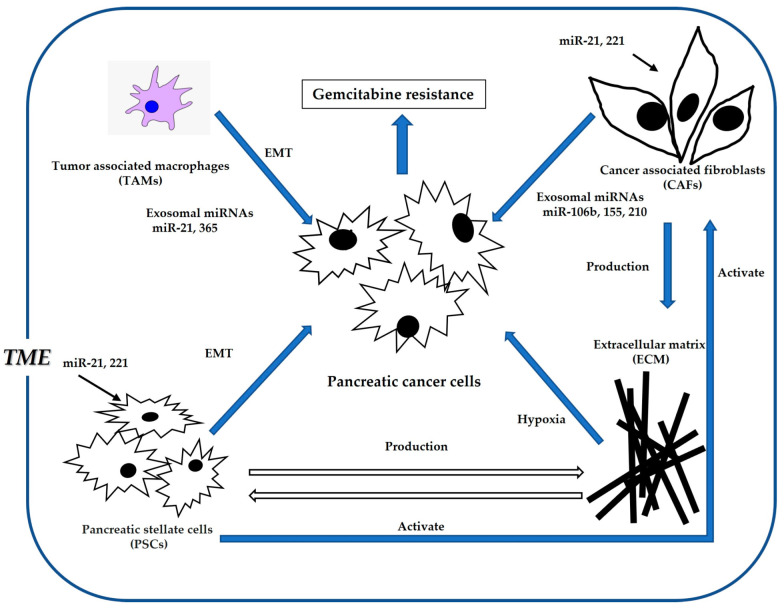
Association of microRNAs with TME (tumor microenvironment) and gemcitabine resistance. TME works to the advantage of cancer cell survival by, for example, inducing gemcitabine resistance and enhancing cell proliferation.

**Figure 3 cancers-15-01230-f003:**
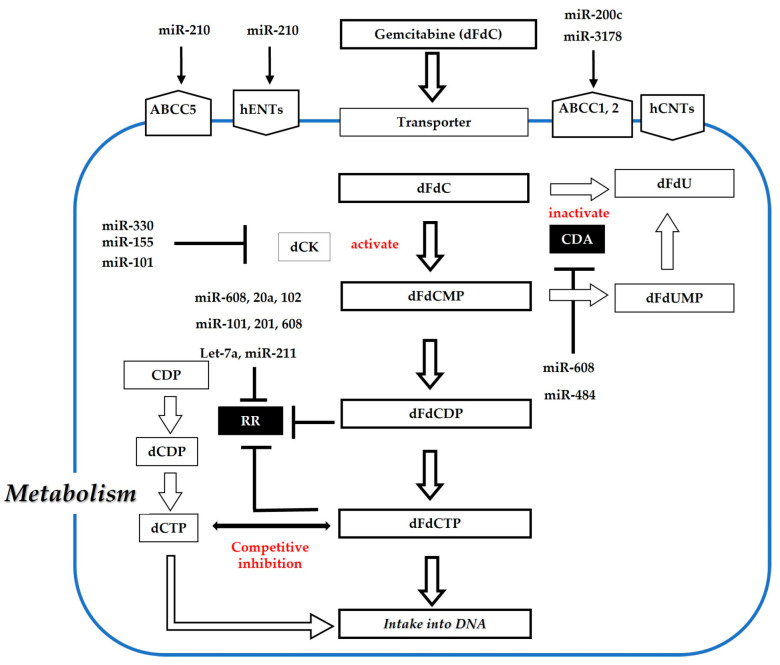
Association of miRNAs with metabolism and gemcitabine resistance. Mainly, transporter and deoxycytidine kinase (dCK) are associated with the activation of gemcitabine. Conversely, cytidine deaminase (CDA) and ribonucleotide reductase (RR) are associated with the inactivation of gemcitabine. These transporters and enzymes are controlled by miRNAs, which results in gemcitabine resistance in pancreatic cancer.

**Table 3 cancers-15-01230-t003:** miRNAs for metabolism.

Author	Ref. Number	miRNA	Target Gene
Sensitivity			
Bhutia YD	[184]	let-7a	*RRM2*
Lu H	[186]	miR-20a	*RRM2*
Fan P	[183]	miR-101	*RRM1*
Amponsah PS	[178]	miR-210	*ABCC5*
Maftouh M	[17]	miR-211	*RRM2*
Rajabpour A	[185]	miR-608	*CDA, RRM1*
Gu J	[179]	miR-3178	*ABC* transporter
Resistance			
Wu Y	[181]	miR-93	*PTEN*
Patel GK	[187]	miR-155	*dCK*
Zhan T	[182]	miR-331	*ST7L*
Xin X	[133]	miR-519c	*ABCG2*

Ref: reference.

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
