# Peer review of "microRNAs Associated with Gemcitabine Resistance via EMT, TME, and Drug Metabolism in Pancreatic Cancer"

_cancers, 2023, doi:10.3390/cancers15041230_

Round 1
Reviewer 1 Report
Cancers-2186914
microRNAs Associated with Gemcitabine Resistance in Pancreatic Cancer
Summary
In the current review, the authors summarize mechanisms that underly gemcitabine resistance that are associated with aberrantly expressed miRNAs in pancreatic cancer. They also discuss how these changes are associated with epithelial-to-mesenchymal transition, the tumor microenvironment, and metabolism. The following comments need to be addressed to enhance the impact of the review article.
Comments:
1. The authors should provide a list of abbreviation used in the review.
2. The authors added a material methods section and described the rationale for choosing the articles which satisfied some distinct criteria in that section. This can be included as part of the introduction.
3. The authors need to discuss miRNA-506, as miR-506 was shown to be epigenetically silenced in pancreatic cancer and restoration of miR-506 suppressed pancreatic cell proliferation, enhanced apoptosis and increased chemosensitivity.
4. The authors need to discuss the role of miRNAs in exosomes and how they confer chemoresistance.
5. The authors need to expand the miRNAs in Gemcitabine resistance section (section 4.2) as title says microRNAs Associated with Gemcitabine Resistance in Pancreatic Cancer or they need to change the title. The authors missed several reported miRNAs involved in chemoresistance including miR-217, miR-92, miR-181, miR-3662 and others.
Author Response
Comments:
- The authors should provide a list of abbreviation used in the review. ➡ Thank you for your comment. I added the list of abbreviations in front of the references.
- The authors added a material methods section and described the rationale for choosing the articles which satisfied some distinct criteria in that section. This can be included as part of the introduction.
I added criteria for choosing articles in the introduction instead of material and methods.
- The authors need to discuss miRNA-506, as miR-506 was shown to be epigenetically silenced in pancreatic cancer and restoration of miR-506 suppressed pancreatic cell proliferation, enhanced apoptosis and increased chemosensitivity.
I appreciate your suggestion. I added miRNA-506 to the table and discussed this in the discussion.
- The authors need to discuss the role of miRNAs in exosomes and how they confer chemoresistance.
As you said, exosome is also a relatively new topic. But this time, I focused on EMT, TME, and metabolism although exosomes are associated with those. So, I added comments for exosomes to the conclusion as a future vision using some references.
- The authors need to expand the miRNAs in Gemcitabine resistance section (section 4.2) as title says microRNAs Associated with Gemcitabine Resistance in Pancreatic Cancer or they need to change the title. The authors missed several reported miRNAs involved in chemoresistance including miR-217, miR-92, miR-181, miR-3662 and others.
Thank you for your advice. I understood what you meant. So, I changed our title. Thus, we removed miR-217, 92, and 181 because of no relation with EMT, TME, and gemcitabine metabolism in pancreatic cancer. Regarding miR-3662, I already added this to the TME session in table 2.
Reviewer 2 Report
The authors present a summary on the role of microRNAs in pancreatic cancer resistance to gemcitabine, our current standard of care for the disease. The review is well structured and succinctly summarise the field as well as the current limitations t therapeutic intervention and the potential for future development of novel therapeutics. I particularly liked how the review was broken into known inherent signalling changes in the cancer cells, tumour microenvironment-derived changes and metabolism related changes that contribute to gemcitabine resistance, then details how microRNAs control each of these aspects. The only changes I would recommend is to include brief figure legends for the summary figures throughout the manuscript – it is not always clear from the figures alone as to how each microRNA is regulating the downstream target i.e. is it upregulation or downregulation?
Author Response
The only changes I would recommend is to include brief figure legends for the summary figures throughout the manuscript.
Thank you for your suggestion. As you said, I added short legends in each figure.
Round 2
Reviewer 1 Report
cancers-2186914-peer-review-v2
microRNAs Associated with Gemcitabine Resistance via EMT, 2 TME and Drug Metabolism in Pancreatic Cancer
The authors addressed all the comments satisfactorily, only one minor point need to be added before acceptance.
The authors were suggested to discuss miRNA-506, as miR-506 was shown to be epigenetically silenced in pancreatic cancer and restoration of miR-506 suppressed pancreatic cell proliferation, enhanced apoptosis and increased chemosensitivity.
The authors added one sentence and one reference regarding this, which needs to be expanded a bit more. The authors can add the following relevant reference and expand the discussion.
Downregulated miR-506 expression facilitates pancreatic cancer progression and chemoresistance via SPHK1/Akt/NF-κB signaling. Oncogene. 2016 Oct 20;35(42):5501-5514. doi: 10.1038/onc.2016.90. Epub 2016 Apr 11.
Author Response
The authors added one sentence and one reference regarding this, which needs to be expanded a bit more. The authors can add the following relevant reference and expand the discussion.
Downregulated miR-506 expression facilitates pancreatic cancer progression and chemoresistance via SPHK1/Akt/NF-κB signaling. Oncogene. 2016 Oct 20;35(42):5501-5514. doi: 10.1038/onc.2016.90. Epub 2016 Apr 11.
Thank you for your suggestion. I added your recommended reference to our discussion.